

# Crossmodal congruency effect scores decrease with repeat test exposure

Daniel Blustein[1,*], Satinder Gill[2,*], Adam Wilson[3] and Jon Sensinger[2,3]

[1] Department of Psychology; Neuroscience Program, Rhodes College, Memphis, TN, United States of America
[2] Institute of Biomedical Engineering, University of New Brunswick, Fredericton, New Brunswick, Canada
[3] Department of Electrical and Computer Engineering, University of New Brunswick, Fredericton, New Brunswick, Canada
[*] These authors contributed equally to this work.

## ABSTRACT

The incorporation of feedback into a person's body schema is well established. The crossmodal congruency task (CCT) is used to objectively quantify incorporation without being susceptible to experimenter biases. This visual-tactile interference task is used to calculate the crossmodal congruency effect (CCE) score as a difference in response time between incongruent and congruent trials. Here we show that this metric is susceptible to a learning effect that causes attenuation of the CCE score due to repeated task exposure sessions. We demonstrate that this learning effect is persistent, even after a 6 month hiatus in testing. Two mitigation strategies are proposed: 1. Only use CCE scores that are taken after learning has stabilized, or 2. Use a modified CCT protocol that decreases the task exposure time. We show that the modified and shortened CCT protocol, which may be required to meet time or logistical constraints in laboratory or clinical settings, reduced the impact of the learning effect on CCT results. Importantly, the CCE scores from the modified protocol were not significantly more variable than results obtained with the original protocol. This study highlights the importance of considering exposure time to the CCT when designing experiments and suggests two mitigation strategies to improve the utility of this psychophysical assessment.

## INTRODUCTION

There is an increasing interest concerning how the human brain represents the space surrounding its body due to converging findings from several different disciplines. The crossmodal congruency task (CCT) is a visual-tactile interference task that has been used to investigate multisensory representation of space in humans (*Spence, Pavani & Driver, 1998*; *Spence, Pavani & Driver, 2004*; *Spence et al., 2004*), including those with brain damage (*Spence et al., 2001a*). Investigation of crossmodal selective attention has been used to demonstrate the detrimental effects of age on the ability to ignore irrelevant sensory information when attending to relevant sensory information (*Poliakoff et al., 2006*). Other studies have investigated changes in the representation of peripersonal space that are elicited by the prolonged use of hand-held tools (*Maravita et al., 2002*; *Holmes & Spence, 2004*; *Holmes, Calvert & Spence, 2007*). Further studies used this task to extend

Corresponding author
Daniel Blustein,
blusteinNeuro@gmail.com

findings regarding physical and pointing tools to virtual robotic tools using techniques from haptics and virtual reality (*Sengül et al., 2012*; *Sengül et al., 2013*). Recently it has been used in conjunction with the rubber hand illusion paradigm to investigate the degree of incorporation of a rubber hand into a person's body schema (i.e., how strongly the rubber hand is experienced as one's own hand) (*Zopf, Savage & Williams, 2010*; *Zopf, Savage & Williams, 2013*).

Although the CCT assessment has become a well-established psychophysics method used to quantify the degree of feedback incorporation in a variety of contexts, in this study we focus on visuo-tactile integration, motivated by its application in the field of neuroprosthetics (*Spence, 2015*). In such investigations, a CCT typically consists of a participant holding two foam blocks in either hand with vibrotactile targets and visual distractors embedded in the top and bottom of each foam block. A trial consists of a random and independent presentation of a single vibrotactile target paired with a single visual distractor from any of the four possible pairs of locations. Participants are instructed to make a speeded response regarding the elevation of the vibrotactile target (i.e., "up", at the index finger verses "down", at the thumb), while simultaneously ignoring visual distractors (*Spence et al., 2001a*).

The CCT is a simple stereotypical behavioral task that provides a robust performance metric of feedback incorporation. Participants are typically slower at discriminating the elevation of vibrotactile targets when visual distractors are presented from an incongruent elevation (i.e., when vibrotactile target and visual distractors are presented from different elevations) as compared to a congruent elevation. The crossmodal congruency effect (CCE) score is calculated as the difference in the reaction time between incongruent and congruent trials and is used as a quantitative performance metric for multisensory representation of space. It has been demonstrated that CCE scores are typically higher when the spatial separation between visual distractor and vibrotactile targets is low (e.g., both locations on same block vs. locations on two different blocks held in different hands) (*Maravita, Spence & Driver, 2003*; *Spence, Pavani & Driver, 2004*).

Despite the widespread use of the crossmodal congruency task in a variety of experimental paradigms, important questions related to task exposure times remain unanswered. For example, in a majority of the studies the experimental protocol consists of multiple blocks of trials with experimental sessions lasting up to approximately 60 min. These hour-long time-intensive experimental sessions might result in extended learning of the task and adversely affect a participant's CCE score. There is also no specific mention of participant selection criteria based on previous knowledge of the CCT. To our knowledge, there exist no studies to date that have investigated modulation of CCE score due to repeated task exposures.

Various neuropsychological and cognitive assessments have shown profound learning effects with repeat exposures, including various reaction time tasks (*Collie et al., 2003*), cognitive function tasks such as the Paced Auditory Serial Addition Test (*Beglinger et al., 2005*), and other interference tasks such as the Stroop Color and Word Test (*Davidson, Zacks & Williams, 2003*; *Beglinger et al., 2005*). If a task learning or practice effect exists for the CCT, results from previous studies that overlooked such an effect may be inaccurate.

Participants tested with previous CCT experience would be expected to have lower CCE scores than participants naïve to the assessment.

Any study that presents more than one CCT exposure to a participant may lead to results affected by a learning effect. One previous study repeatedly ran the CCT with a single subject, Patient J.W. (*Spence et al., 2001b*). *Spence et al. (2001b)* only directly compared performance within counterbalanced sessions in order to reduce the effect of differential motivation or fatigue. However, there is no discussion of a task learning effect, the results from five different sessions across two days are presented side-by-side (*Spence et al., 2001b*), and the same patient J.W. appears in another CCT study (*Spence et al., 2001a*). Additionally, studies utilizing a counterbalanced within-subjects design may average out the learning effect, such as *Holmes, Calvert & Spence (2007)* in which each participant completed six blocks of 96 trials each across three different counterbalanced conditions. A learning effect, if present, would conflate the research findings that form the basis of our current understanding of multisensory space representation in humans. In the present study we show evidence that CCE scores decrease with repeated task exposure sessions and that this learning effect is persistent over time. We propose two mitigation strategies, including a shortened CCT protocol that we compare to the established protocol. Best practices for future CCE score use are suggested.

# EXPERIMENT 1

We first sought to characterize any change in CCE scores and performance with repeat exposure to the psychophysics protocol.

## Materials and methods
### Participants
Twelve healthy volunteer participants were recruited from the local community (11 male, 1 female; age mean $\pm$ SD $= 23.4 \pm 6.6$ years; 11 right hand dominant, 1 reported no particular hand dominance). All participants had normal or corrected to normal vision, no disorder of touch, were able to use both foot pedals, and were naïve to the CCT without previous exposure to or knowledge of the task. Participants were informed about the general purpose of the research and were given the opportunity to ask questions but they were not aware of the specific goals of the research. Participants received no compensation. Experiments were conducted under human ethics approval of the US Department of the Navy's Human Research Protection Program and the University of New Brunswick (Fredericton, NB, Canada) Research Ethics Board (Protocol #2016-032). Written informed consent was obtained from each participant before conducting experiments and no compensation was provided to participants.

### Materials and apparatus
The experimental test platform was designed around the National Instruments (NI) myRIO embedded hardware system to achieve millisecond-timing accuracy. The three digital outputs of the myRIO system were used to drive one 3 mm green LED, termed the fixation LED, and two 3 mm green LEDs named distractor LEDs. Two 308-107 Pico Vibe

8 mm vibratory motors from Precision Microdrives were used as the vibrotactile targets for the thumb and index finger. The analog outputs of the myRIO system were used to drive STMicroelectronics L272 power operational amplifiers that drove the vibratory motors. The amplifiers were necessary to provide sufficient driving current to the vibratory motors. Two OnStage KSP100 keyboard sustain pedals were interfaced to the myRIO system through digital inputs to measure a participant's speeded responses to the vibrotactile targets. NI LabVIEW was used to develop the firmware for the myRIO system to randomly activate the visual distractors and vibrotactile targets at a fixed interval and measure the speeded response time. A desktop NI LabVIEW graphical user interface (GUI) was designed to interact with the myRIO embedded system, allowing the experimenter to set various experimental parameters such as vibratory stimulus amplitude, the number of trials and to specify the filename to record the timing results of speeded responses.

### Experiment design

In Experiment 1 we assessed the learning effect over five sessions of the CCT. Each of the twelve participants, naïve to the CCT, completed five consecutive days of CCT testing using the standard 8-block protocol (*Spence, Pavani & Driver, 2004*). The number of sessions and subjects was determined with a power analysis using preliminary CCT results from eight pilot subjects (see Appendix S1). For each subject, testing was scheduled at the same time each day for all five visits. The testing time across participants was variable to accommodate different individual schedules.

### CCT implementation

During CCT testing, participants sat comfortably on a chair in front of a table in a dimly illuminated room. They wore over-ear noise-canceling headphones playing Brownian noise to mask background noise. Identical vibrotactile stimulation motors were placed on the thumb and index fingertips of the participant's right hand (see the 'Materials and Apparatus' section above for hardware details). Distractor light emitting diodes (LEDs) were placed on the thumb and index finger of the participant's right hand, while a fixation LED was centered between the distractor LEDs using a plastic mounting strip. Figure 1 shows the system setup used for this experiment.

At the beginning of the CCT assessment, participants were instructed to make a speeded response to vibrotactile targets that were presented randomly to either the thumb or index fingertip. Participants responded by pressing the left or right foot pedals to indicate stimulation of the thumb or index finger, respectively. Participants were explicitly instructed to respond as fast as possible, while making as few errors as possible. Participants were also informed that visual distractors would be presented simultaneously with the vibrotactile targets but that they were completely irrelevant to the vibrotactile target discrimination task. They were specifically instructed to ignore visual distractors by keeping their eyes open and fixating their vision on the central fixation LED. The fixation LED was presented 1,000 ms before the vibrotactile targets and visual distractors were simultaneously presented for 250 ms.

The CCT test session was preceded by a practice session consisting of three blocks of 10 trials each. The first block of the practice session consisted of presenting the fixation LED

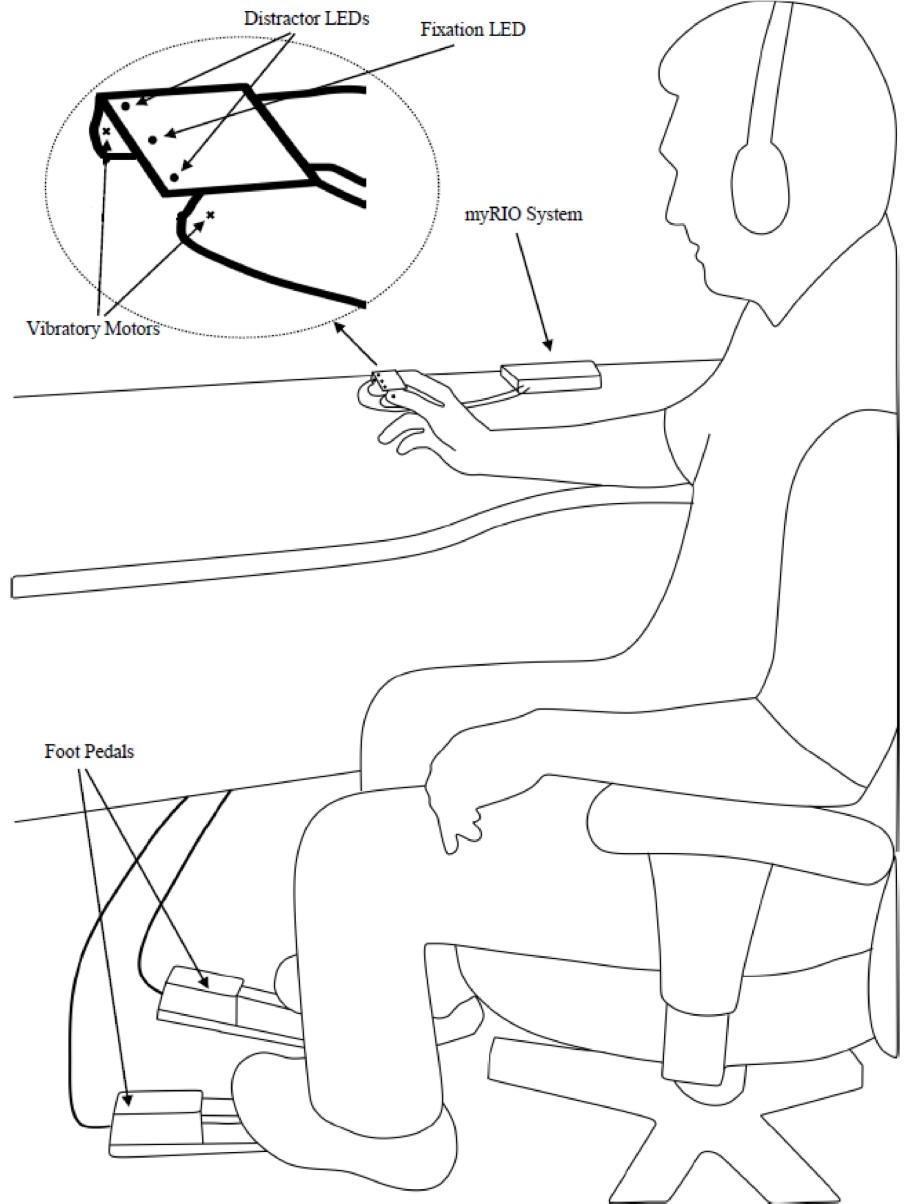

**Figure 1  Overall system setup.** Subjects make speeded responses with foot pedals to vibratory feedback presented on the index finger or thumb. Subject is asked to focus on the fixation LED while ignoring the distractor LEDs. See text for method details.

and vibrotactile targets only (i.e., no visual distractors were presented) so that participants could familiarize themselves with the vibrotactile target discrimination. We calculated a generalized reaction time for each participant as the mean of the last 5 trials of the first practice block. The visual distractors were presented along with vibrotactile targets in the second and third blocks of the practice session. This was followed by the experimental session during which 64 trials were presented to each participant during each block of

trials. Each trial ended when the participant responded by pressing one of the foot pedals or no response was made within 1,500 ms of target onset.

All CCT sessions consisted of eight blocks of 64 trials each. Each block lasted for approximately four minutes and a break period of two minutes was provided between consecutive blocks.

### Analysis

All analyses were conducted using custom scripts in MATLAB software (2017 release, The MathWorks, Inc., Natick, MA, USA). Practice session trials were not analyzed except to calculate generalized reaction time. To calculate the CCE score from the test trials, we first discarded trials with an incorrect response, trials with a premature response (i.e., reaction time less than 200 ms), and trials with a delayed response (i.e., reaction time greater than 1,500 ms) as these conditions most likely occurred due to lapses in attention (*Spence, Pavani & Driver, 2004*; *Sengül et al., 2012*). The remaining trials in each block were used to calculate the mean congruent and mean incongruent reaction times. The CCE score for each block was calculated by taking the difference between mean incongruent and congruent reaction times. The mean of the block CCE scores was used to calculate the CCE score for each participant for a particular exposure session. Selection error rates were used as a separate metric of analysis. We evaluated attenuation across sessions using a one-way repeated-measures ANOVA. All data are available on Dryad (10.5061/dryad.150v8g3).

## Results—Experiment 1: CCE score decreases over repeated exposures

We measured the CCE score of 12 subjects across five sessions using the conventional CCT method of 8-blocks. We observed a significant effect of exposure number on CCE score determined with a repeated-measures ANOVA with Greenhouse-Geisser correction (Fig. 2) ($F(2.00, 21.95) = 6.93$, $p = .005$, partial-eta $= .39$). Tests of within-subjects polynomial contrasts over session number indicated a significant linear trend ($F(1, 11) = 8.63$, $p = .013$), a significant 4th order trend ($F(1, 11) = 5.14$, $p = .044$), and a quadratic trend ($F(1, 11) = 4.83$, $p = .05$). Post-hoc comparisons with Bonferroni adjustments showed no significant pair-wise differences. The statistically-supported trends match the observed initial decrease in CCE score and subsequent stabilization, indicating a task learning effect (Fig. 2).

To verify that the change in CCE score was due to a learning effect and not attributed to other interacting factors such as variability in motivation or baseline reactivity, we analyzed the CCT selection error rates, generalized reaction times measured from practice trials, and overall reaction times for congruent and incongruent stimuli. We found no significant effect of exposure number on CCT correct trial rate (Fig. 3A), as indicated by a repeated-measures ANOVA with Greenhouse-Geisser correction ($F(1.15, 12.66) = 0.53$, $p = .51$). The correct trial rates on congruent and incongruent trials independently did not show consistent trends (Fig. S1). Repeated-measures ANOVA with Greenhouse-Geisser correction indicated a significant effect of exposure on both incongruent reaction time ($F(2.16, 23.78) = 13.57$, $p < .01$) and congruent reaction time ($F(1.46, 16.07) = 8.89$, $p < .01$) (Fig. 3B). Although mean generalized reaction time during the practice trials

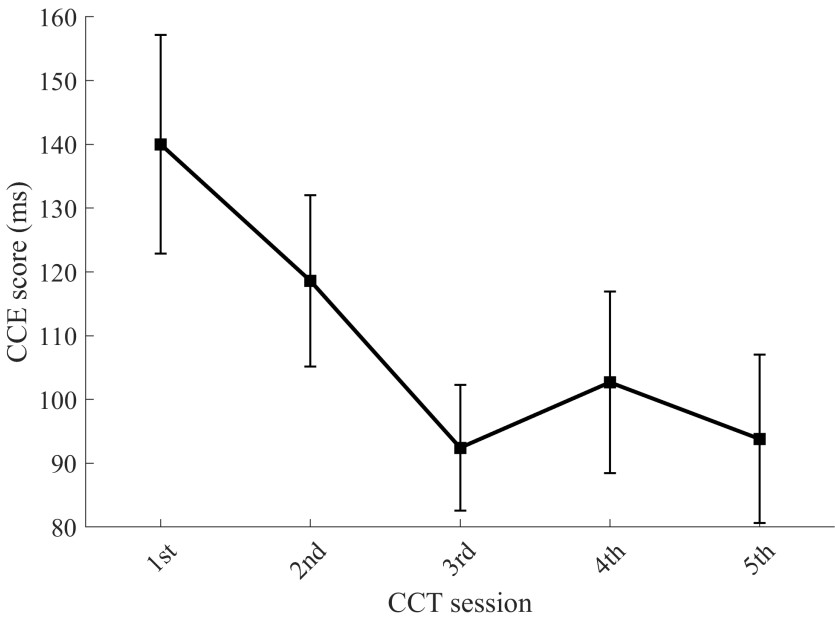

**Figure 2** **CCE score decreases with task exposure.** Each data point represents the mean CCE score for 12 subjects. Error bars show standard error. Results from Experiment 1.

decreased across each session, this trend was not statistically significant as indicated by a repeated-measures ANOVA with Greenhouse-Geisser correction ($F(2.62, 28.81) = 2.59$, $p = .079$) (Fig. S2). To summarize, the only metrics presenting similar exposure-dependent decreases as the CCE scores (Fig. 2) were the congruent and incongruent reaction times (Fig. 3B).

## EXPERIMENT 2

After observing a statistically significant CCT learning effect in both incongruent and congruent reaction times but not in error rates and generalized reaction times in Experiment 1, we wanted to see if the learning effect persisted over longer time periods. In Experiment 2, we assessed the persistence of the CCT learning effect using a within-subjects design comparing the responses of subjects tested in Experiment 1 with their CCE results following the same protocol about six months later.

### Materials and methods

All twelve participants from Experiment 1 were asked to return for follow-up testing on a volunteer basis after six months with no CCT exposure. Eight return participants completed Experiment 2 (8 male; age mean $\pm$ SD $= 22.3 \pm 7.0$ years; 7 right hand dominant, 1 reported no particular hand dominance). All experimental parameters were the same as in Experiment 1 including the CCT testing apparatus, practice sessions, and test protocol including numbers of blocks and trials. We calculated CCE scores in the way described for Experiment 1. We also ran an intraclass correlation coefficient analysis [two-way mixed model with single measurements] and calculated the standard error of measurement [SEM
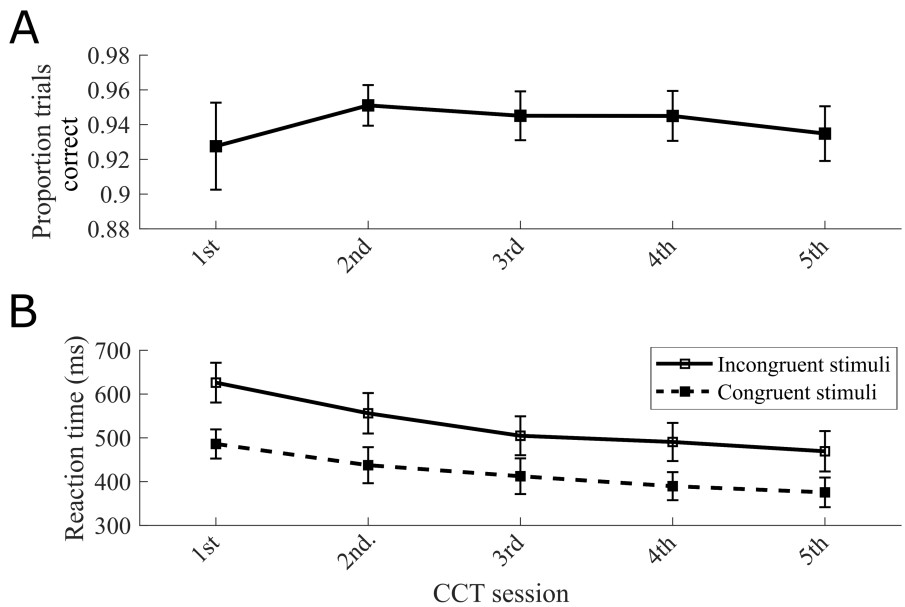

**Figure 3** **Decreasing CCE score trend matched in isolated reaction times but not in CCT correct trial rate.** (A) Plot of proportion correctly selected trials over repeated exposures. An incorrect trial occurred when the wrong pedal was pressed (e.g., vibration stimulus on the thumb resulted in subject pressing the pedal indicating an index finger stimulus). (B) Congruent and incongruent reaction times plotted independently. Results from Experiment 1.

$= \text{SD} \sqrt{(1 - \text{ICC})}$] to assess the test-retest reliability between the 6 month follow-up visit and the Day 5 visit of Experiment 1 (*Weir, 2005*).

## Results—Experiment 2: CCT learning effect persists over time

Re-visit CCE scores were similar, and even slightly lower, than the CCE scores measured on Session 5 of Experiment 1, at least 6 months prior (mean time between sessions ± SD = 201.6 ± 9.1 days; range = 191–209 days) (Fig. 4). Similarity between Session 5 and the re-visit session was indicated by a high intraclass correlation coefficient (ICC(3,1) = .71) and a low standard error of measurement (SEM = 21.6 ms) compared to baseline reliability data calculated by comparing Session 1 and re-visit results (ICC(3,1) = .46; SEM = 45.7 ms).

## EXPERIMENT 3

The persistence of the learning effect observed in Experiment 2 suggests that CCE scores should only be considered after task learning has stabilized. However, in certain research or clinical settings with subject access or time constraints, extended testing may be impractical. In Experiment 3, we sought to explore mitigation strategies to diminish the impact of the CCT learning effect on research results. Specifically, we tested a modified CCT protocol designed to reduce task exposure to determine if the shortened testing could: 1. Produce valid CCT results; and 2. Mitigate the learning effect.

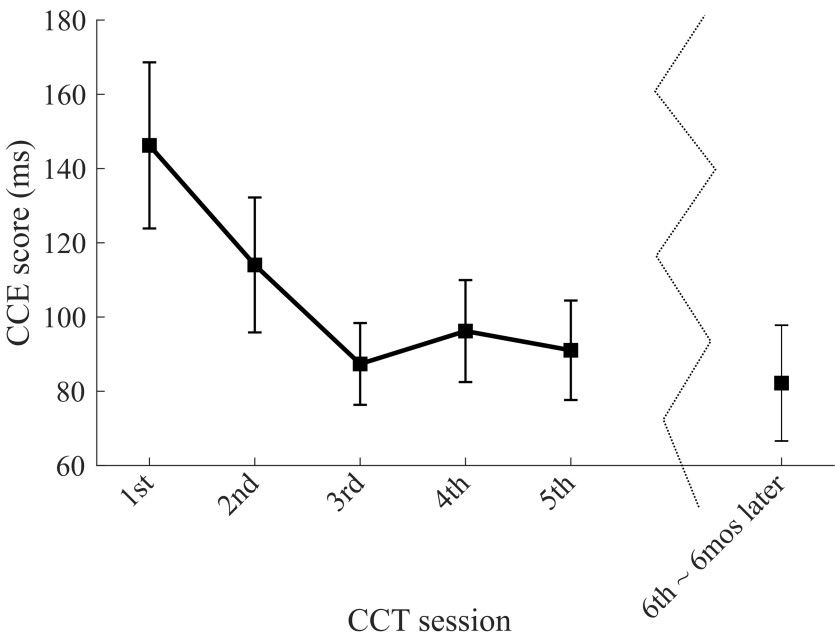

**Figure 4** **Persistent CCT learning effect after 6-month follow-up testing.** Eight subjects from Experiment 1 returned for a sixth visit about 6 months after their initial visits. Data shown are means and standard error for those eight subjects. Results from Experiment 2.

## Materials and methods

We recruited eighteen volunteer subjects for Experiment 3, all of whom were naïve to the CCT (and had not participated in Experiments 1 and 2). Half of the subjects ($n = 9$) were randomly assigned to test with a shortened CCT protocol (3 female; aged 24–57 years, mean $\pm$ SD = 33.8 $\pm$ 10.4 years; 8 right hand dominant, 1 unspecified hand dominance) and the other nine subjects tested under the standard protocol used in Experiment 1 and 2 (3 female; aged 20–32 years, mean $\pm$ SD = 26.7 $\pm$ 4.6 years; 8 right hand dominant, 1 left hand dominant).

The testing apparatus was identical to Experiments 1 and 2. In the standard protocol, as described above in the CCT Implementation section, participants completed eight separate testing blocks within each CCT session. In the shortened protocol, we reduced each CCT session to four blocks, with each block still comprised of 64 trials. All other specifications of the CCT implementation were kept consistent with Experiments 1 and 2. We expected fewer CCT trials (in the four block protocol) to result in a reduced learning effect, and we sought to determine if CCE scores would maintain their consistency with the shortened protocol.

Participants in Experiment 3 completed two sessions of the CCT, one on each day for two consecutive days. Each participant tested with the same protocol on each day (standard or shortened) and testing sessions were scheduled for each individual at the same time both days. CCE scores were calculated the same way as in Experiments 1 and 2. In Experiment 3 we ran paired $t$-tests to compare results between the standard and shortened protocol.

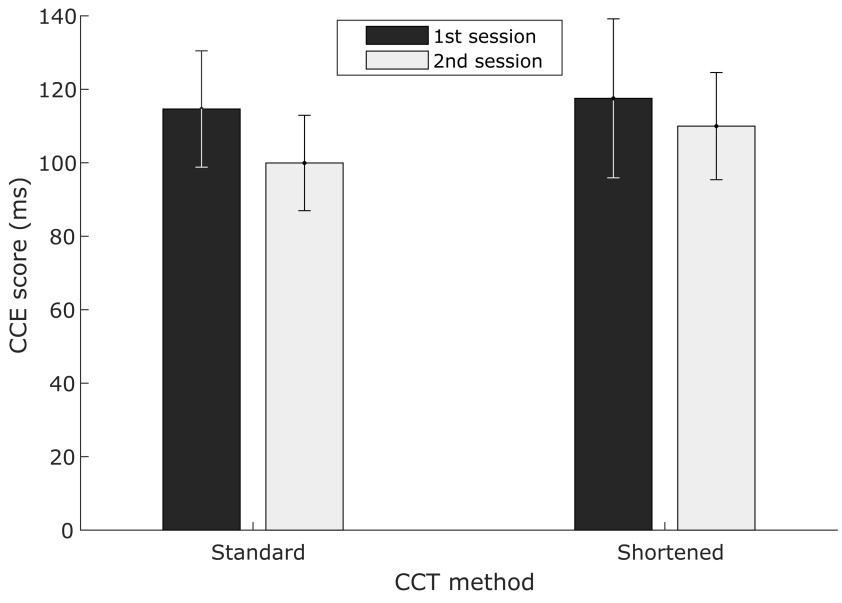

**Figure 5  First exposure vs. second exposure mean CCE scores for standard and shortened protocols.**
Means and standard error for 8-block standard ($n = 9$) and 4-block shortened ($n = 9$) protocol partici­pants. Dark-colored bars represent the first exposure session and light-colored bars represent the second exposure session. Results from Experiment 3.

## Results—Experiment 3: modified CCT protocol reduces learning effect

The shortened protocol reduced the duration of task exposure in each session from eight blocks to four blocks. In the 4-block protocol, mean CCE score dropped from 117.5 ms in the first session to 110.0 ms in the second session (Fig. 5). This was a smaller reduction than observed with the standard 8-block protocol: 114.6 ms in the first session dropped to 99.9 ms in the second session (Fig. 5). For the 8-block protocol, mean CCE score dropped by 14.7 ms from the first exposure to the second, compared to a 7.6 ms drop for the 4-block protocol. Variability in the second session was only slightly higher for the 4-block protocol (SD = 43.8 ms) compared to the 8-block protocol (SD = 39.0 ms). The variability difference was more pronounced when analyzing the first session results [4-block SD = 64.9 ms; 8-block SD = 47.5 ms]. None of these differences were statistically significant (paired $t$-tests, $p > .05$), however.

## DISCUSSION

The CCT is a well-established quantitative measure of feedback incorporation that has important implications for objective human sensorimotor assessments (*Spence, Pavani & Driver, 1998*; *Spence, Pavani & Driver, 2004*; *Spence et al., 2004*; *Spence, 2015*). The resulting scores have been used to quantify the rubber hand illusion (*Zopf, Savage & Williams, 2010*) and show promise for assessing neuroprosthetic devices (*Spence, 2015*). Many applications of the CCT require repeated testing—for example, testing the same subject under different sensory feedback conditions to see which one enables the highest degree of feedback

incorporation. Oftentimes in research or clinical settings time allotted for sensorimotor assessment can be limited. This study sought to characterize CCT performance through repeat testing to inform its future implementation.

In Experiment 1, we found that with repeat testing, CCE scores decreased over the initial exposure sessions. Different motivation levels or strategies fail to explain the CCE score trend as CCT error rates did not show a trend matching the decreasing scores. The decreasing trend in generalized reaction times, although not statistically significant, would likely affect both incongruent and congruent reaction times similarly. Thus, we consider the decrease in CCE score to be evidence of a task learning effect. Interestingly, we observed no significant pair-wise differences between CCE scores on different days, even though our statistical tests showed overall significant effects. This was likely due to the large number of pairwise comparisons and the conservative Bonferroni correction we applied.

Although other psychophysical paradigms similarly show significant practice effects (*Collie et al., 2003*; *Davidson, Zacks & Williams, 2003*; *Beglinger et al., 2005*), for the case of the CCT, scores tend to increase with increased training durations (*Blustein, Wilson & Sensinger, 2018*). The observation of a learning effect in light of these counteracting processes suggests that either an increase in CCE score does not occur with repeat assessment (rather than generalized training), or the learning effect has a much greater impact in the resulting CCE score.

In Experiment 2, we demonstrated that the learning effect is persistent as CCE scores after six months remained near the stabilized scores observed at the end of five days of consecutive testing. The persistence of the learning effect provides an opportunity for researchers to only use CCE scores collected after learning has stabilized.

In certain circumstances, additional testing to arrive at stable CCE scores may not be feasible. In Experiment 3, we showed that a shortened CCT protocol with four testing blocks instead of eight reduced the impact of the learning effect across two testing sessions. Experiment 3 was underpowered statistically and thus these findings can only be considered as preliminary trends. In certain circumstances where repeat testing is required but time constraints exclude the possibility of testing until learning asymptotes, the reduced exposure protocol may be necessary. Importantly, CCE score variability did not substantially increase with the 4-block protocol, although the statistical caveats apply here as well.

Reducing the number of blocks per session has the added benefit of reducing fatigue, improving the ability of subjects to concentrate over the duration of the study, and shortening the length of the session to approximately 30 min (which in turn makes it viable to test a greater number of subjects or for use in a clinical setting). Using a modified version of the CCT with only four blocks of 64 trials has many potential benefits, with the only drawback identified being a slight increase in the variability of results.

Through this characterization of the learning effect associated with the CCT, we have provided support for two distinct mitigation strategies. If extended testing is feasible, researchers can provide enough practice so that CCE scores are only measured and compared after task learning has stabilized. This would be of particular use when tracking an individual's progress over time with repeat testing. In situations where reduced testing is required, researchers can use the 4-block CCT protocol to reduce overall task
exposure and subsequently reduce the impact of the learning effect on the results. It is important that these strategies not be mixed; CCE scores should only be compared to other scores that were collected in the same manner. Randomizing condition order to provide counterbalanced groups can also serve to reduce the learning effect's impact on results and can be useful in observing group differences (*Holmes, Calvert & Spence, 2007*). However, this counterbalancing approach will effectively average out the learning effect, which would lead to reduced magnitudes of any observed differences between conditions. It is not obvious what is being 'learned' through repeated task exposure. We suspect that subjects strengthen an internal model of the foot pedal location tied to the vibration stimulus, resulting in faster response times. Anecdotal support for this idea comes from one subject's comment: "When the feedback is in the opposite location to the light, I sometimes find myself thinking about which pedal to press only to realize I've already pressed the correct pedal". Somatosensory reorganization of an individual's body schema through training may also be implicated (*Cardinali et al., 2009*).

The initial session CCE scores (i.e., before learning effect stabilization) appear to be higher than scores reported by other researchers (*Spence, Pavani & Driver, 2004*; *Zopf, Savage & Williams, 2010*). One potential explanation for this observation is that other studies may have different baseline CCE scores due to the experimental inquiry, such as studies using spatial misalignment (*Spence, Pavani & Driver, 2004*) or artificial hands (*Zopf, Savage & Williams, 2010*; *Marini et al., 2014*). We would expect a lower degree of feedback incorporation, and thus a lower CCE score (*Maravita et al., 2002*), under these conditions compared to the ideal conditions we used with both vibratory feedback and visual distractors aligned in place on one of the subject's hands. Additionally, different experimental set-ups could account for differences in CCE scores. Others have recorded user inputs differently, for example with a rocking heel-toe foot pedal setup (*Spence, Pavani & Driver, 2004*) or with finger presses (*Zopf, Savage & Williams, 2010*). Other variable techniques include the exclusion of trials with extensive eye movement (over 9% omitted in one study (*Spence, Pavani & Driver, 2004*)), or the use of no-go trials as a control (*Zopf, Savage & Williams, 2010*). Furthermore, differences in the duration and parameters of practice or familiarization trials before testing could lead to differences in the resulting CCE scores. In a hypothetical experimental setup with all other factors held constant, this study would suggest that a longer practice phase could result in a more pronounced learning effect before testing began, resulting in a lower CCE score. It might also be helpful to explore more granular trial-to-trial variation in the CCT results using mixed effects modeling.

Another possible explanation for the observed variability in CCE scores across studies is the different methods used to apply feedback. Different feedback modalities result in different CCE scores (*Mayer et al., 2009*; *Frings & Spence, 2010*). Even when comparing studies using vibration feedback, the method of application can vary and may result in CCE score differences. We used small vibratory motors (0.8 cm diameter vibrating surface) but others have used larger bone conduction vibrators (1.6 cm × 2.4 cm vibrating surface, *Spence, Pavani & Driver, 2004*) or small speakers (0.9 cm diameter vibrating surface, *Zopf, Savage & Williams, 2010*), all could result in different degrees of feedback incorporation. To

focus on the learning effect, we kept all testing parameters in this study constant. Although we expect these results to be similar across other CCT implementations, the magnitude of the learning effect with different CCT parameters is unknown.

Differences in subject characteristics could also explain differences in CCE scores. All of the subjects in this study were initially naïve to the CCT, but it is unclear if that was an inclusion criteria in other studies. Even comparing the first session results from this study across the two cohorts of subjects tested with 8 CCT test blocks we see differences in mean CCE score. In Experiment 1, a mean CCE score of 140 ms was observed (Fig. 2) compared to 115 ms in Experiment 3 (Fig. 5). More careful monitoring of additional potentially confounding variables may be helpful. To summarize, there are lots of factors that could affect CCE score and it is impossible to determine if results reported elsewhere are capturing pre-, mid- or post-learning effect scores.

To our knowledge, this is the first study that has demonstrated the modulation of CCE score due to task overexposure. We recognize some limitations with this study that could warrant additional studies. Although we collected a generalized reaction time from practice trials, more sophisticated tracking of mental and physical fatigue would be helpful to better inform our understanding of the observed learning effect. Although CCT testing for individual participants was conducted at the same time of the day for each exposure, no strict guidelines were used to ensure that participants were in similar mental or physical states before each test session. Nevertheless, the learning effect we have described has important implications for future use of the CCT. Going forward, the learning effect must be considered when measuring CCE scores by only using scores from after learning has stabilized, or by using a modified protocol that reduces exposure to the task.

## CONCLUSION

This study identified a learning effect with an established psychophysics metric: repeat exposure to the crossmodal congruency effect task resulted in reduced scores. The effect persisted during follow-up testing after a six-month hiatus. This learning effect must be considered when using the CCT. For limited testing, we have presented a modified protocol with reduced trial blocks that can be used to limit task exposure. For ongoing testing where multiple tests are necessary, we suggest researchers allow this learning effect stabilize before relying on CCE scores for comparisons. Consideration of the learning effect is important to properly contextualize CCT results.

## ACKNOWLEDGEMENTS

We thank Matthew Sampson for assistance in data collection and Noah Mesa for assistance in data analysis.

### Funding

This work was funded by the Defense Advanced Research Projects Agency (DARPA) through the HAPTIX program (award number N66001-15-C-4015) and by the National Institutes of Health (award number 7R01NS081710-02). There was no additional external funding received for this study. The funders had no role in study design, data collection and analysis, decision to publish, or preparation of the manuscript.

### Grant Disclosures

The following grant information was disclosed by the authors:
Defense Advanced Research Projects Agency (DARPA) through the HAPTIX program: N66001-15-C-4015.
National Institutes of Health: 7R01NS081710-02.

### Competing Interests

The authors declare there are no competing interests.

### Author Contributions

- Daniel Blustein conceived and designed the experiments, analyzed the data, contributed reagents/materials/analysis tools, prepared figures and/or tables, authored or reviewed drafts of the paper, approved the final draft, supervised research assistant.
- Satinder Gill conceived and designed the experiments, performed the experiments, analyzed the data, contributed reagents/materials/analysis tools, prepared figures and/or tables, authored or reviewed drafts of the paper, approved the final draft.
- Adam Wilson conceived and designed the experiments, performed the experiments, analyzed the data, contributed reagents/materials/analysis tools, approved the final draft.
- Jon Sensinger conceived and designed the experiments, approved the final draft, provided mentorship and project oversight.

### Human Ethics

The following information was supplied relating to ethical approvals (i.e., approving body and any reference numbers):

The University of New Brunswick Research Ethics Board and the U.S. Department of the Navy's Human Research Protection Program granted ethical approval to carry out this study (UNB REB #2016-032, DoD IRB Protocol #2015-033).

### Data Availability

The raw data is available in Dryad: Data from: Crossmodal congruency effect scores decrease with repeat test exposure doi: 10.5061/dryad.150v8g3.

## Supplemental Information

Supplemental information for this article can be found online at http://dx.doi.org/10.7717/peerj.6976#supplemental-information.

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
