# Peer review of "Crossmodal congruency effect scores decrease with repeat test exposure"

_PeerJ, doi:10.7717/peerj.6976_

## Round 0.1 · original submission · Minor Revisions

Thank you very much for your submission to PeerJ. Two reviewers have kindly reviewed your article and provided feedback to you.

One concern that both reviewers raised was related to the fact that you only considered and tested one class of crossmodal correspondence. While they suggest that should have tested other classes, I do not believe that such a requirement is necessary for the acceptance of your article to PeerJ, however you might consider highlighting this limitation to your conclusions more explicitly.

I will not reiterate the reviewers' other comments here but I encourage you to consider them all carefully and respond to each one as you prepare your revised article for resubmission.

Reviewer 1 ·

Basic reporting

Generally solid, if selective review of huge literature.

Would do better to compare the decline in magnitude of this effect with what one sees in other paradigms

Experimental design

Solid, borrowing well-established paradigm

Validity of the findings

I would have liked to see a more granular analysis of decline of CCE across trials.......

Additional comments

Review.
I guess my major issue with this manuscript would that the results may not surprise anyone. Most behavioural effects decline with practice.
These results would be more interesting/striking in the context of other results where interference effects declined more or did not decline at all for contrast.
In this regard, I seem to remember a paper from c. 20 years ago on object-based inhibition of return, that some authors found and other authors couldn’t replicate. The key turned out to be that the object based effect was only there for one block. One side of debate was researchers at University of Bangor from memory. Maybe that would be helpful to cite.
Plus, I guess it is useful to identify the magnitude of reduction in CCE over trials/blocks, but one other simple means of eliminating its influence is perfectly randomizing condition order, no? Not always possible, and not always done, but feels like the majority of studies may have counterbalanced /randomized order in which diff trial types presented so possibly eliminating problem, at least in terms of interpreting results. Surprisingly, this isn’t mentioned as a solution in abstract and elsewhere
The authors might also want to discuss other factors that have been shown to influence magnitude of CCE effect too? And perhaps consider which component of behavioural effect they think is/are declining: see Shore, D. I., Barnes, M. E., & Spence, C. (2006). The temporal evolution of the crossmodal congruency effect. Neuroscience Letters, 392, 96-100.
Perhaps also Shore, D., & Simic, N. (2005). Integration of visual and tactile stimuli: Top-down influences require time. Experimental Brain Research, 166, 509-517.
Is this a learning effect, a practice effect? Is there a meaningful difference between these terms

“42 representation of space in humans (Spence, Nicholls, Gillespie & Driver 1998;” From memory Spence et al 1998 was an exogenous spatial cuing study, and CCT only properly introduced by Pavani et al. 2000), or 1998 Psychonomics Society presentation by C Spence.

76 “time-intensive experimental sessions….” Meaning what exactly?

“77 score. Also,” Try not to start sentence with also…

78 “knowledge, there exist no studies to date
79 that have investigated modulation of CCE score due to repeated task exposures. If a task learning
80 effect exists for the CCE, results from previous studies that ignored such an effect may be
81 misinterpreted. Participants tested with previous CCE experience would be expected”
I guess here you could mention repeated testing of patient JW by Spence et al. 2001a, 2001b, as an example where this issue might be problematic?
Spence, C., Shore, D. I., Gazzaniga, M. S., Soto-Faraco, S., & Kingstone, A. (2001). Failure to remap visuotactile space across the midline in the split-brain. Canadian Journal of Experimental Psychology, 55, 54-61.


One thing that might deserve more mention is that most CCT experiments actually have a number of practice trials/blocks before proper data collection starts. Hence this should be factored in to any concern about practice/learning effects swaying result, and this point may be relevant to discussion about why diff studies seem to report diff magnitude effects.

“All participants had normal or corrected to 122 normal vision, no disorder of touch, were able to use both foot pedals, and were naïve to the CCE
123 task.” In what sense naïve, they had never heard of task, they had never performed it, or both?

Often CCE shows up in error rates, and in the patient studies just mentioned inverse efficiency (combining RT and accuracy into a single performance measure) was used. How does learning/practice influence errors rates seems an important question that is worth discussing in perhaps more detail, and also considering in review of other studies.

Given that the CCT has been used in many more studies than those reported here, perhaps it is worth stressing a little more that you are just mentioning a selective review of studies that have used the task?

Note inconsistent article capitalization between articles:

“355 Sengül, A., van Elk, M., Rognini, G., Aspell, J.E., Bleuler, H., & Blanke, O. (2012). Extending 356 the body to virtual tools using a robotic surgical interface: evidence from the crossmodal 357 congruency task. PLoS ONE, 7(12), e49473. doi:10.1371/journal.pone.0049473
358 Spence, C. (2015). The Cognitive Neuroscience of Incorporation: Body Image Adjustment and 359 Neuroprosthetics.

Why is volume in brackets?
“380 Zopf, R., Savage, G., & Williams, M.A. (2013). The Crossmodal Congruency Task as a Means 381 to Obtain an Objective Behavioral Measure in the Rubber Hand Illusion Paradigm.382 Journal of Visualized Experiments, (77), e50530.

A further review of CCT and findings =
Spence, C., Pavani, F., Maravita, A., & Holmes, N. P. (2008). Multi-sensory interactions. In M. C. Lin & M. A. Otaduy (Eds.), Haptic rendering: Foundations, algorithms, and applications (pp. 21-52). Wellesley, MA: AK Peters.

I didn’t find the x axes of figures talking about blocks and experiments very helpful. Surely what reader wants to know is how many trials does it take eg to reach steady performance.

Shading between bars in Fig 5 could be made much more distinct.

·

Basic reporting

Gill et al. reported across 3 experiments to demonstrate that crossmodal congruency effect is subject to modulation due to (over-)exposure/learning effects in a visual-tactile interference task. They then suggest that use of CCE scores/protocols would be better off by taking exposure time into consideration.

Experimental design

I found the logic for including Exp 3 to be unclear. My understanding is that CCE will decrease as the subjects are exposed to more sessions (more trials), as it shows from Exposure 1 to 5 (see main results in Figure 1). It came with no surprise that fewer practice in the 4-block condition will produce a smaller CCE learning effect since the participants in the 4-block condition have had fewer trials for them to reach their asymptote than those participants in the 8-block condition. Therefore, the only valid comparison is to have both groups of participants to receive same numbers of trials in both 4-block (double number of trials per block) and 8-block conditions. Otherwise this suggested mitigation procedure is pretty invalid. All it says with Experiment 3 is that we should give the participants as little practice/exposure as possible but then if it is the case then it goes against the main argument/proposal of the manuscript that we should use CCE data only after the subjects reach their CCE asymptote.

Validity of the findings

The second criticism is that this study covers only 1 aspect of CCE where in the crossmodal literature there are several other classes crossmodal correspondence (e.g., visual-audio among others). So I pose the question to the authors that if they can provide more evidence to demonstrate some kind of “generalization of this CCE to other crossmodal correspondence domains”? This generalization across domains is theoretically very important to instantiate their arguments. Only if the authors can demonstrate (or refute) such domain-generality in CCE I would deem this paper to be worthy to be published in PeerJ. I suggest the authors pick a dataset from a domain (I suggest visual-audio correspondence) and look into whether the CCE will stabilize across exposure sessions to test for any analogous learning effect.

Additional comments

Overall the analyses are fairly elementary that only mean session-specific RTs are compared. Note that there has been a surge of interest in the research of trial-history in relation to current behaviors (e.g., Carlos Brody's recent Nature paper (https://www.nature.com/articles/nature25510; and also Merav Ahissar’s 2012 and 2014 papers in PLoS computational biology). Specifically, whether a preceding congruent trial will pose a differential effect on current congruent trial (also for an incongruent trial on a current incongruent trial, likewise for the other two possible combinations). It would be theoretical interesting to examine the trial-by-trial influence in each of the sessions using GLM analysis incorporating various (sets of) regressors.

Reviewer 3 ·

Basic reporting

See comments

Experimental design

See comments

Validity of the findings

See comments

Additional comments

The authors investigate the effect of repeated exposure on CCE by presenting the task for five consecutive days. This could induce some kind of learning that decreases the CCE scores and that persists over six months. Moreover, the study shows the authors' effort to find a shortened length procedure of the task to mitigate possible participants’ fatigues and learning effects. Therefore, the overall aim of this study seems worth asking. However, my major concerns are about their conclusions and interpretations. Specifically, in my opinion the authors should be more careful and they cannot state that “…results from previous studies that ignored such an effect may be misinterpreted” and “A learning effect that has been ignored by the field may render inaccurate our current understanding of multisensory space representation in humans”. Indeed, to the best of my knowledge and based on the literature that they presented on the introduction, they have been the first that have presented the task for five consecutive days. The learning effect showed by their participants, after five exposures, could be not present in participants that have been exposed to a single session. So, the decrease in the CCE scores that they found, could be not relevant, or even not present, in the previous studies. Broadly speaking, I think that authors should enrich their introduction and should improve their experimental design (i.e. by using a more precise measure for the overall reaction time; or by measuring the overall reaction time before and after the sessions; or by measuring a possible learning/fatigue effect during a single session of 8 blocks; or by measuring if a learning effect persists after six months in participants tested with the short protocol).
Now, I will list several issues in more detail:
Abstract
Line 14: It seems that there is a confusion between the “effect” and the “task” labels. In my opinion, Crossmodal congruency task is the procedure and the implementation of the design (well described in the introduction, line 56-61) that has been used to quantify the Crossmodal congruency effect (CCE) score, namely, the difference between incongruent and congruent trials.
Introduction
In my opinion, authors should enrich the introduction by adding more studies that have investigated the Crossmodal congruency effect, especially studies that have used a repeated exposure procedure.
Line 39: The authors should introduce and explain the topic of “multisensory representation of human space” and incorporation of “information into body schema”.
Line 54: the first time that authors introduce the CCE they should explain what it is.
Line 56: the acronym CCE stands for Crossmodal congruency Effect, not for the task. To avoid ambiguity authors should consider to refer to the effect like “CCE” and to the task like “CCT”. It could be less confounding for the reader.
Line 65: CCE score is calculated by the difference between incongruent and congruent trials, not the reverse.
Line 70: CCE is the effect, not the task.
Line 78 - 85: The authors investigate the effect of repeated exposure by presenting the task for five consecutive days (line 78-79). This could induce some kind of learning that decreases the CCE scores and that persists over six months. To the best of my knowledge and on the basis of literature presented in the introduction, no one has exposed participants to a consecutive days procedure. Authors should explain better this point and should add some references about studies that have used a repeated exposure protocol, if they exist. Otherwise, they cannot state a possible misinterpretation of the CCE from the previous studies about the multisensory space representational (line 81-83). My point is that eight blocks or 60 minutes of a single session, might not imply a learning effect at all. Authors, to the best of my understanding, did not explore a possible learning effect within the 8 blocks.
Line 119: Authors should present the three experiments separately. That is, Experiment 1: participants (number, mean age and SD, left/right hand dominance), materials, experimental design/procedure, analysis and results. The same structure for experiments 2 and 3. CCE test implementation (line 90) and materials and apparatus (line 131), that are in common between the three experiments can be placed together in the description of experiment 1.
Line 150: “…CCE task..” I suggest to use a specific acronym for the task.
Line 155: Authors state that they have used a “rule drop task” to measure generalized reaction time. Authors should indicate a reference, and explain better why the need to measure the generalized reaction time.
Moreover, they should explain why they use the ruler drop task, usually used with children or adults with clinical disability, rather than another more common and, probably, more precise task, like a “simple rt task” (see, Elliot R., 1968 Simple visual and simple auditory reaction time: a comparison).

Line 160 and 165: design of exp 2 and exp 3 should be described after experiment 1 in their specific section.
Line 166: I wonder if 18 participants are sufficient to explore the exposure and learning effect. In the power analysis reported for experiment 1 is specified that with 2 sessions authors should need 45 participants. So, why they tested only 18 participants, furthermore divided into 2 groups? If the power analysis conducted for experiment 1 was not appropriate for experiment 3, why authors did not conduct a power analysis specifically for exp 3? Authors should disentangle this point.
Line 181-185: Analysis of exp2 and exp3 should be placed on their specific area.
Line 192: which specific exposure sessions differ from each other? Authors should run a post-hoc analysis to determine it.
Results
Exp 1
Line 198-199: what kind of “interacting factors” could intervene? Please provide examples.
Line 200: in figure 3 it seems that authors represent proportion correct instead of error rate. Please check the data. Moreover: why do authors do not represent the proportion correct divided for congruent and incongruent stimuli, like they do with the RT? I wonder if a learning effect should not emerge also with the participants’ hit responses, inducing participant to have pretty good performance also with incongruent stimuli and weakening the CCE scores. If a learning effect exists, authors should find some effect also in the hit responses, otherwise one could speculate that participants simply became faster. Authors should introduce these data and eventually explain the absence/presence of modifications in the hit responses.
Line 202 to 204: authors should present the post-hoc analysis to highlight which data point is different from the others. Once again, in figure 3b it seems that the participants are faster but the difference from incongruent and congruent seems to remain significant. So the CCE seems to persist over the learning effect.
Discussion
Line 250: Authors should explain better how they can exclude the “Fatigue effect”. Indeed, overall reaction times were always measured before the task. I think that, to evaluate the fatigue effect, authors should have measured the overall reaction times before and after the task, in each session.
Line 251: If authors had wanted to match the overall reaction times with the decrease in the CCE scores, probably they should have used a similar kind of response, i.e. authors could have used a simple reaction time task with a foot responses to measure overall reaction times task. Indeed, it could be that in five consecutive exposure the participants could have become faster with foot responses but not with their hand responses.
Line 251: in line 155 authors refer to “ruler drop task” here, in line 251, they refer to “ruler grasp task”. Please check for consistency.
Line 257: “the persistence of the learning effect… has stabilized”. My doubt persists: are there some studies that have used a consecutive five days procedure? Otherwise, authors should have had evaluated a possible learning effect after the eight blocks of the single procedure. Please, add some studies in the introduction which use a consecutive days procedure and, in the discussion, compare them with your results.
Line 260 - 261: It would be interesting to evaluate if the persistence over six months of this “learning effect” would manifest itself even after the short procedure.
Line 291 -304: “Additionally, different experimental setups could account for differences in the CCE scores… feedback incorporation”. If CCE scores are so sensitive to experimental setups, feedback modalities, etc., why authors decide to use these different experimental setups compared to other studies?
Line 299: please double check there is a typo in the reference.
Line 311: “… to determine if results reported elsewhere…” please add some references, results reported from which studies?
Figure 1: If, on one hand, I appreciate the effort to recreate the design of the overall system setup, on the other hand, I think that a real picture of the device on the hand could be more explicative to understand the implementation. So, I suggest adding a picture to enrich the understanding of the procedure.
Figures 2, 3 and 4
In my opinion, labels in the x axes should be “1st session”, “2nd session”, etc… and not “exp.”since “exp.” might be misinterpreted as “experiment” and not for “exposure”.
Figure 2 and 3: in the caption authors should state that results come from experiment 1.
Figure 3a: please check the data, it seems that authors report proportion correct instead of error rate.
Figure 4: in the caption authors should state that they are CCE scores from experiment 2.
Figure 5: in the caption authors should state that results come experiment 3.

---

## Round 0.2 · Minor Revisions

Thank you very much for submitting your revised article to PeerJ. I believe that you have adequately responded to each of the reviewers' suggested edits and questions and I do not feel the need to send this article out for review again.

However, in my own reading of your article, I noticed a few areas (particularly your methods) that required additional clarification. I attach an annotated copy of your article with my suggested edits and questions.

In particular, I think you need to provide more detailed information about the protocols used for each of the three experiments. One way to do this, as suggested by one of the reviewers, is to present each protocol and associated results separately. However, if you wish to keep your current formatting, you may do so, but additional information is still required (please see my comments in the attached). PeerJ does not have a specific formatting requirement, and we allow flexibility to you to determine the best way to present your protocols and data, but I do encourage to you ensure that each of the three methods are described in a such a way that they can be replicated (I am not certain that the current description is clear enough to facilitate this).

Once you have attended to these final outstanding suggestions, it will be my pleasure to accept your article for publication in PeerJ.

---

## Round 0.3 · accepted · Accept

Thank you very much for responding to my suggested revisions. You have clearly thoroughly revised the structure of your article, increasing its clarity. It is my pleasure to accept your article for publication.

#